# The Effects of Angiotensin II or Angiotensin 1-7 on Rat Pial Microcirculation during Hypoperfusion and Reperfusion Injury: Role of Redox Stress

**DOI:** 10.3390/biom11121861

**Published:** 2021-12-10

**Authors:** Dominga Lapi, Maurizio Cammalleri, Massimo Dal Monte, Martina Di Maro, Mariarosaria Santillo, Anna Belfiore, Gilda Nasti, Simona Damiano, Rossella Trio, Martina Chiurazzi, Barbara De Conno, Nicola Serao, Paolo Mondola, Antonio Colantuoni, Bruna Guida

**Affiliations:** 1Department of Biology, University of Pisa, Via San Zeno, 31, 56127 Pisa, Italy; maurizio.cammalleri@unipi.it (M.C.); massimo.dalmonte@unipi.it (M.D.M.); 2Department of Clinical Medicine and Surgery, Federico II University of Naples, Via S. Pansini, 5, 80131 Naples, Italy; martinadimaro@hotmail.it (M.D.M.); mariarosaria.santillo@unina.it (M.S.); anna.belfiore@unina.it (A.B.); gilda.nasti@unina.it (G.N.); simona.damiano@unina.it (S.D.); rostrio@unina.it (R.T.); martina.chiurazzi@unina.it (M.C.); barbara.deconno@gmail.com (B.D.C.); nicolaserao75@gmail.com (N.S.); paolo.mondola@unina.it (P.M.); antonio.colantuoni@unina.it (A.C.); bruna.guida@unina.it (B.G.)

**Keywords:** cerebral microcirculation, renin–angiotensin system, hypoperfusion–reperfusion, angiotensin II, angiotensin-1-7

## Abstract

Renin–angiotensin systems produce angiotensin II (Ang II) and angiotensin 1-7 (Ang 1-7), which are able to induce opposite effects on circulation. This study in vivo assessed the effects induced by Ang II or Ang 1-7 on rat pial microcirculation during hypoperfusion–reperfusion, clarifying the mechanisms causing the imbalance between Ang II and Ang 1-7. The fluorescence microscopy was used to quantify the microvascular parameters. Hypoperfusion and reperfusion caused vasoconstriction, disruption of blood–brain barrier, reduction of capillary perfusion and an increase in reactive oxygen species production. Rats treated with Ang II showed exacerbated microvascular damage with stronger vasoconstriction compared to hypoperfused rats, a further increase in leakage, higher decrease in capillary perfusion and marker oxidative stress. Candesartan cilexetil (specific Ang II type 1 receptor (AT_1_R) antagonist) administration prior to Ang II prevented the effects induced by Ang II, blunting the hypoperfusion–reperfusion injury. Ang 1-7 or ACE2 activator administration, preserved the pial microcirculation from hypoperfusion–reperfusion damage. These effects of Ang 1-7 were blunted by a Mas (Mas oncogene-encoded protein) receptor antagonist, while Ang II type 2 receptor antagonists did not affect Ang 1-7-induced changes. In conclusion, Ang II and Ang 1-7 triggered different mechanisms through AT_1_R or MAS receptors able to affect cerebral microvascular injury.

## 1. Introduction

Cerebral circulation has been extensively studied under the pathophysiological conditions of ischemia and reperfusion; in particular, previous data obtained using in vitro models indicate that angiotensin II (Ang II) affects cerebral microcirculation through the activation of angiotensin II Type 1 receptor (AT_1_R) during ischemia–reperfusion injury [1,2,3,4]. The renin–angiotensin–system (RAS) can fine tune the amount of Ang II acting on microvascular networks. Renin facilitates the release of angiotensin I (Ang I) from angiotensinogen, a peptide released by the liver. On the other hand, Ang I can be processed by the classical angiotensin converting enzyme (ACE1), thus leading to the formation of Ang II, a molecule that causes arteriolar constriction and other microvascular effects through AT_1_R [5]. However, Ang II can be processed by ACE2, a transmembrane metallopeptidase constituted 805 amino acids, cloned by Tipnis and coworkers in 2000 [6]. This enzyme removes a single residue from Ang II to generate Angiotensin 1-7 (Ang 1-7) and a single residue from Ang I to yield Angiotensin 1–9, which is further processed to generate Ang 1-7. Regardless of how it is produced, Ang 1-7 determines arteriolar dilation and effects opposite to those triggered by Ang II, acting through the MAS (Mas oncogene encoded protein) receptor [7,8,9]. The interplay between ACE1 and ACE2 is fundamental to facilitate the balance of constriction and dilation induced, respectively, by Ang II and Ang 1-7 [10,11], a balance that is known to modulate the microvascular functions in all organs of living organisms.

Under conditions of cerebral ischemia–reperfusion, it has been observed that AT_1_R antagonism may reduce the infarct size in both normotensive and hypertensive rats [12], indicating that Ang II could play a role in the mechanisms linked to cerebral damage, where the predominance of Ang II in Ang 1-7 activity may trigger arteriolar constriction and an increase in arteriolar resistance. Ang 1-7 has been seen to restore vascular reactivity and endothelial cell function, mainly in experiments on isolated large vessels and in cell cultures [13,14].

Therefore, it is crucial to undertake an in vivo study of the effects of angiotensin II and angiotensin 1-7 on cerebral microcirculation in conditions of ischemia–reperfusion and to assess the mechanisms involved. To this aim, we evaluated the pial microcirculation in rats that underwent a protocol of hypoperfusion–reperfusion in the absence or in the presence of different pharmacological treatments applied prior to hypoperfusion and at the beginning of reperfusion either through topical application or intravenous injection. In particular, rats were treated with Ang II alone or in the presence of the AT_1_R antagonist candesartan cilexetil in order to verify whether the effect of Ang II was through AT_1_R, as previously demonstrated in vitro [1,2,3,4,15]. Moreover, rats were administered angiotensin 1-7 alone and in the presence of Mas receptor antagonist (A779) or Ang II type 2 receptor (AT_2_R) antagonist (PD123319) to clarify the intracellular mechanisms triggered by angiotensin 1-7 [16]. In addition, in a similar experimental set, we assessed the role of ACE2 in the balance between Ang II and Ang 1-7 through the administration of diminazene aceturate, an ACE2 activator. In the different experimental conditions, we evaluated arteriolar diameter changes, microvascular leakage, formation of reactive oxygen species (ROS), adhesion of leukocytes to venular walls and capillary perfusion.

## 2. Materials and Methods

### 2.1. Experimental Design

According to the Guide for the Care and Use of Laboratory Animals published by the US National Institutes of Health (NIH Publication 8th edition, 2011) and to institutional rules for the care and handling of experimental animals, with approval of the protocol from the University of Pisa Ethical Committee (Protocol No. 156/2017-PR), the experiments were carried out on Wistar rats, weighing 250–300 g. Animals were randomly assigned to 18 experimental groups. The first group was sham-operated (S group, n = 5), The second group (Hc group, n = 11) comprised rats subjected to bilateral common carotid artery occlusion (BCCAO) for 30 min (hypoperfusion) and to 60 min of reperfusion. The other sixteen groups comprised animals submitted to BCCAO and reperfusion and to pharmacological treatments either with local application on pial surface or right jugular vein infusion. Groups and treatments were reported in Table 1.

In particular, Groups 3–10 were constituted by male rats subjected to BCCAO and reperfusion and treated with local application on pial surface for 10 min before hypoperfusion and for 10 min at the beginning of reperfusion of following compounds: Group 3 (AngII-L group; n = 11) were treated with Ang II (5 × 10^−9^ M); Group 4 (Ang1-7-L group; n = 11) with Ang 1-7 (10^−9^ M); Group 5 (C-L group; n = 11) with candesartan cilexetil (2 × 10^−7^ M); Group 6 (C-AngII-L group; n = 11) with candesartan cilexetil (2 × 10^−7^ M) plus Ang II (5 × 10^−9^ M); Group 7 (A-Ang1-7, n = 11) with Mas receptor antagonist (A779) (10^−3^ M) plus Ang 1-7; Group 8 (P-Ang1-7, n = 11) with AT_2_R antagonist (PD123319) (10^−3^ M) plus Ang 1-7; Group 9 (D-L group; n = 11) with diminazene aceturate (8 × 10^−6^M); Group 10 (D-AngII-L group; n = 11) with diminazene aceturate (8 × 10^−6^M) plus Ang II (5 × 10^−9^M).

Groups 11-18, male rats subjected to BCCAO and to reperfusion, were treated with the following compounds infused in the right jugular vein 10 min before hypoperfusion and at the beginning of reperfusion: Group 11 (AngII-J group; n = 11) were administered with Ang II (10 µg/kg); Group 12 (Ang1-7-J; n = 11) with Ang 1-7 (1 mg/kg); Group 13 (C-J; n = 11) with candesartan cilexetil (15 mg/kg); Group 14 (C-AngII-J group; n = 11) with candesartan cilexetil (15 mg/kg) plus Ang II (10 µg/kg); Group 15 (A-Ang1-7, n = 11) with A779 (1 mg/kg) plus Ang 1-7; Group 16 (P-Ang1-7, n = 11) with PD123319 (5 mg/Kg) plus Ang 1-7; Group 17 (D-J group, n = 11) with diminazene aceturate (10 mg/kg); Group 18 (D-AngII-J group, n = 11) with diminazene aceturate (10 mg/kg) plus Ang II (10 µg/kg).

The last 4 groups comprised female rats subjected to BCCAO and to reperfusion and treated: Group 19; AngII-LF group, n = 11, with local application on pial surface of AngII at 5 × 10^−9^ M; (Group 20; Ang1-7-LF group; n = 11) or of Ang 1-7 at 10^−9^ M. Group 21 rats, AngII-JF group; n = 11, were administered AngII at 10 µg/kg or Ang 1-7 at 1 mg/kg in the right jugular vein (Group 22; Ang1-7-JF, n = 11).

In Groups 2-22, five animals were employed for microvascular studies, and six animals were used to determine the in vivo ROS formation by 2′-7′-dichloro-fluorescein-diacetate (DCFH-DA) assay after hypoperfusion (n = 3) or after reperfusion (n = 3).

### 2.2. Treatments

The solutions used for local administration were dissolved in artificial cerebrospinal fluid (aCFS) [16]. Composition of aCSF was: 119 × 10^−3^ M NaCl, 2.5 × 10^−3^ M KCl, 1.3 × 10^−3^ M MgSO_4_·7H_2_O, 1 × 10^−3^ M NaH_2_PO_4_, 26.2 × 10^−3^ M NaHCO_3_, 2.5 × 10^−3^ M CaCl_2_ and 11 × 10^−3^ M glucose (equilibrated with 10.0% O_2_, 6.0% CO_2_ and 84.0% N_2_; pH 7.38 ± 0.02). The temperature was maintained at 37.0 ± 0.5 °C with a water bath.

For jugular vein infusion, Ang II, Ang 1-7, A779, PD123319 and diminazene aceturate were dissolved in sterile saline [17,18,19,20,21,22,23,24,25,26]. Candesartan cilexetil was dissolved in dimethyl sulfoxide [22,23]. It was then diluted in aCSF before administration [21]. The final concentration of dimethyl sulfoxide (0.33%) did not affect the pial microcirculation, as evaluated in preliminary experiments.

Ang II or Ang-1-7 were applied for 10 min prior to hypoperfusion and for 10 min at the beginning of reperfusion. Their application lasted 5 min (local application by superfusion) or 3 min (intravenous infusion). Candesartan cilexetil, A779, PD123319 and diminazene aceturate were administered 10 min before the local or intravenous application of Ang II. The dosages of drugs used were chosen according to previous studies [19,20,21,22,23,24,25,26,27] and through preliminary experiments aimed to evaluate the different microvascular effects.

Drugs applied topically together with the aCSF were superfused at 0.5 mL/min by a peristaltic pump. During superfusion the intracranial pressure was maintained at 5 ± 1 mmHg and measured by a pressure transducer (Gould Windograf recorder: model 13-6615-10S, Gould, OH, USA) connected to a computer. Drugs intravenously administered were infused at a rate of 0.1 mL/min. The drugs were purchased from Merck Life Science, Milan, Italy.

### 2.3. Surgical Animal Preparation

Under anesthesia induced with α-chloralose (50 mg/kg, intraperitoneal) and maintained with supplemental injections of α-chloralose (30 mg/kg, i.v., every hour), animals were tracheotomized and mechanically ventilated [28,29,30,31]. The two common carotid arteries were isolated and successively clamped for 30 min to induce hypoperfusion. The femoral vein and artery were catheterized to administer supplemental anesthetic dose and fluorescent tracers. Fluorescein isothiocyanate (FITC) bound to dextran (molecular weight 70 kDa; Merck Life Science) was injected with 500 mg/kg of 5% wt/vol solution to label pial vessels. The baseline values of arteriolar diameter, microvascular permeability, ROS formation or capillary perfusion were recorded 3 min after FITC infusion. Rhodamine 6 G (Merck Life Science) was used at 10 mg/kg in 0.3 mL and administered at the end of reperfusion to label leukocytes. The mean arterial blood pressure (MABP) was recorded by a catheter placed in the femoral artery. Another catheter was inserted into the jugular vein for Ang II infusion. Rats were secured on a special heating stereotaxic frame to maintain a constant body temperature (37.0 ± 0.5 °C), which was being checked through a rectal probe. To visualize the pial microvasculature a closed cranial window was prepared on the left parietal cortex after removing the skull and the dura mater, as previously reported in detail [29].

### 2.4. Pial Microvascular Observation

The observation was performed using a fluorescence microscope (Leitz Orthoplan, Wetzlar, Germany) equipped with long-distance objectives [5×, numerical aperture (NA) 0.12; 10 ×, NA 0.25; 20×, NA 0.40; 40×, NA 0.60], a 10× eyepiece and a filter block (Ploemopak, Leitz) used for FITC and rhodamine [29]. A 100-watt mercury lamp offered the epi-illumination, while to prevent the overheating of the preparations, a heat filter (Leitz KG1) was assembled. A DAGE MTI 1000 low-light-level camera [30,31], connected to a Sony PVM 122 CE monitor and to an imaging computerized system was used to record the images in real time. The recordings were stored through a computer-based frame grabber (Pinnacle DC 10 plus, Avid Technology, Burlington, MA, USA) [32]. MABP and heart rate were recorded by a Gould Windograph through a Statham PD 23 transducer connected to the catheterized artery. For each microvascular network the diameter and length of vessels were evaluated by a computerized method (MIP Image; Institute of Clinical Physiology, CNR, Pisa, Italy), and pial arterioles were classified by Strahler’s scheme [32].

Microvascular permeability, used as an index of blood–brain barrier integrity, was measured by evaluating fluorescent dextran extravasation from venules and expressed as normalized gray levels (NGLs): NGL = (I − Ir)/Ir, where Ir was the baseline gray level at the end of microvasculature filling with fluorescence, and I was the value at the end of hypoperfusion or reperfusion. The MIP image program permitted us to obtain gray levels, averaging data derived from 5 windows, 50 × 50 µm (10× objective) each, located outside the venules. To identify the same regions of interest, a computer-assisted device was used for XY movement of the microscope table. The leukocyte adhesion to the vessel walls over a 30 sec time period was reported as number of adherent cells/100 μm of venular length (v.l.)/30 sec, utilizing appropriate magnification (40× objective). Perfused capillaries were evaluated as the length of the capillaries showing blood flow (PCL), assessed by MIP image in an area of 150 × 150 μm [29]. All data were measurements performed by two blind operators and were compared to avoid bias due to single-operator judgement. In all cases, the results overlapped.

### 2.5. ROS Production Detection

DCFH-DA is an in vivo marker for oxidative stress of cells and tissues previously used in our laboratory [30,33]. DCFH-DA was dissolved in aCSF and superfused over pial surface for 15 min during hypoperfusion or reperfusion, as previously reported in detail [30]. DCF fluorescence intensity is correlated directly with intracellular reactive ROS level, which was evaluated by an appropriate filter (522 nm) and measured by NGL.

### 2.6. Statistical Analysis

The data are reported as mean ± SEM. Kolmogorov–Smirnov test was used to verify if the data were normally distributed. Sequentially, parametric or non-parametric tests were applied according to data distribution. Diameter and length of pial vessels, as well as ROS formation, were analyzed with non-parametric test (Kruskal–Wallis test). The microvascular permeability, leukocyte adhesion and perfused capillary length in the different experimental groups were compared with parametric tests (One-way ANOVA followed by Bonferroni’s post hoc test). The SPSS 14.0 statistical package (IBM Italia, Segrate, MI, Italy) was utilized, and the statistical significance was set at *p* < 0.01.

## 3. Results

All microvascular networks were mapped to classify the pial arterioles in orders, according to the Strahler’s method [31]; each network was characterized by five orders of arterioles, as reported in Table 2. We studied one order 4 arteriole, two order 3 arterioles and two order 2 arterioles during each experiment. The data obtained revealed homogenous responses; therefore, we report only the data concerning the order 3 vessels because these arterioles were more numerous in each experimental preparation and together with the vessels of order 2, they were most responsive to the different treatments [28].

Animals belonging to the S group did not show changes in the baseline values of arteriolar diameter, microvascular permeability, ROS formation or capillary perfusion for the overall period of observation. For this reason, no bars indicating the baseline were reported in any figure.

### 3.1. The Hypoperfusion and the Subsequent Reperfusion Caused Significant Microvascular Alterations

In the animals submitted to hypoperfusion and reperfusion, at the end of hypoperfusion we observed that all arteriolar orders decreased in diameter; in particular, the vessels were constricted by 22.3 ± 1.5% of the baseline (*p* < 0.01 vs. the baseline and S group) (Figure 1). Moreover, both the microvascular leakage (Figure 2) and ROS formation (Figure 3) significantly increased compared with the baseline (NGL = 0.23 ± 0.02 and 0.27 ± 0.03, *p* < 0.01 vs. the baseline and S group, respectively).

At the end of reperfusion, the diameter of all arterioles remained reduced with respect to the S group (−15.7 ± 2.1% of the baseline, *p* < 0.01 vs. the baseline and S group) (Figure 1). The fluorescent dextran leakage was further increased compared to that observed at the end of hypoperfusion (NGL = 0.46 ± 0.03, *p* < 0.01 vs. the baseline and S group) (Figure 2 and Figure 4(A1,A2)). These events were accompanied by a marked ROS formation (NGL = 0.33 ± 0.03, *p* < 0.01 vs. the baseline and S group) (Figure 3), significant leukocyte adhesion to the venular wall (10 ± 1/100 µm v.l./30 s, *p* < 0.01 vs. S group) (Figure 5) and reduced capillary perfusion (PCL = −52 ± 2% of the baseline, *p* < 0.01 vs. the baseline and S group) (Figure 6).

### 3.2. Local Application of Ang II Exacerbated Microvascular Damage Due to Hypoperfusion and Reperfusion

After local application, Ang II did not significantly affect microvascular networks under baseline conditions. The diameter of arterioles, indeed, did not change compared to the baseline. There was no fluorescent leakage along the vessels and no adhesion of leukocytes to the venular wall, and all capillaries were perfused. On the contrary, at the end of hypoperfusion pial arterioles showed a significant reduction in diameter by 28.5 ± 2.2% of the baseline, which was, however, lower than in the Hc group (*p* < 0.01 vs. the baseline, S and Hc groups) (Figure 1). The microvascular permeability was higher than that of the Hc group (NGL = 0.35 ± 0.03, *p* < 0.01 vs. the baseline, S and Hc groups) (Figure 2), while ROS formation did not differ from that of the Hc group (NGL = 0.29 ± 0.04, *p* < 0.01 vs. the baseline and S group) (Figure 3).

At the end of reperfusion vessel diameters did not differ from those of the Hc group (−20.5 ± 1.8% of the baseline, *p* < 0.01 vs. the baseline and S groups) (Figure 1). Microvascular leakage (Figure 2 and Figure 4(B1–B3)) and ROS formation (Figure 3) were more significant than that of the Hc group (NGL = 0.53 ± 0.02 and 0.45 ± 0.02, *p* < 0.01 vs. the baseline, S and Hc groups, respectively). The number of adherent leukocytes was similar to that of the Hc group (12 ± 2/100 µm v.l./30 s, *p* < 0.01 vs. the baseline and S group) (Figure 5). Moreover, the capillary perfusion was more compromised than that of the Hc group (PCL = −64 ± 3% of the baseline, *p* < 0.01 vs. the baseline, S and Hc groups) (Figure 6).

### 3.3. Local Application of Ang 1-7 Prevented Microvascular Damage Due to Hypoperfusion and Reperfusion

After local application, Ang 1-7 did not significantly affect microvascular networks under baseline conditions. In addition, at the end of hypoperfusion pial arteriolar diameters did not significantly change compared to the S group, with Ang 1-7 preventing the reduction elicited by hypoperfusion (diameter decreased by 2.6 ± 1.3% of the baseline, *p* < 0.01 vs. Hc and AngII-L groups) (Figure 1). Both microvascular leakage (Figure 2 and Figure 4(C1,C2)) and ROS formation (Figure 3) were almost completely prevented by Ang 1-7 (NGL = 0.02 ± 0.01 and 0.04 ± 0.01, *p* < 0.01 vs. Hc and AngII- L groups, respectively) (Figure 3).

At the end of reperfusion, the arterioles did not show any significant change in diameter when compared with baseline conditions (diameter increased by 2.3 ± 0.5 % of the baseline, *p* < 0.01 vs. Hc and AngII-L groups) (Figure 1). The microvascular permeability (Figure 2 and Figure 4(C3)) was preserved (NGL = 0.03 ± 0.01, *p* < 0.01 vs. Hc and AngII-L groups), while ROS formation (Figure 3) and the adhesion of leukocytes to the venular walls (Figure 5) were almost completely prevented (NGL = 0.06 ± 0.02 and 2 ± 1/100 µm v.l./30 s, *p* < 0.01 vs. Hc and AngII-L groups, respectively). The capillary perfusion was preserved (PCL = −4.2 ± 1.3 % of the baseline, *p* < 0.01 vs. Hc and AngII-L groups) (Figure 6).

### 3.4. Jugular Vein Infusion of Ang II or Ang 1-7 Achieved Similar Results to Those Triggered by Local Application

The animals subjected to jugular vein infusion with Ang II or Ang 1-7 showed the same microvascular changes as those observed in the groups with local application of either Ang II or Ang 1-7 alone.

Ang II, administered into the jugular vein, did not significantly affect microvascular networks under baseline conditions, while at the end of hypoperfusion the pial arteriolar diameter was reduced by 30.2 ± 1.5% of the baseline (*p* < 0.01 vs. the baseline, S and Hc groups). The microvascular leakage was more marked than in the Hc group (NGL = 0.37 ± 0.02, *p* < 0.01 vs. the baseline, S and Hc groups). The ROS formation was: NGL = 0.27 ± 0.03 (*p* < 0.01 vs. baseline and S group). After 60 min of reperfusion the diameter changes were similar to those observed in the Hc group (−19.8 ± 2.1% of the baseline, *p* < 0.01 vs. baseline and S groups). The microvascular permeability and ROS formation were: NGL = 0.48 ± 0.03 and 0.41 ± 0.03, respectively (*p* < 0.01 vs. the baseline, S and Hc groups). The number of adhered leukocytes was 13 ± 1/100 μm v.l./30 s (*p* < 0.01 vs. the baseline and S group). The capillary perfusion was markedly compromised (−68 ± 4% of the baseline, *p* < 0.01 vs. the baseline, S and Hc groups).

Ang 1-7, administered into the jugular vein, did not modify microvascular networks under baseline conditions. At the end of hypoperfusion no pial arterioles the showed any significant diameter changes compared to the S group (diameter decreased by 2.2 ± 1.5% of the baseline, *p* < 0.01 vs. Hc, AngII:L-J groups). Microvascular leakage and ROS formation were prevented (NGL = 0.04 ± 0.01 and 0.03 ± 0.02, *p* < 0.01 vs. Hc, AngII:L-J groups, respectively). At the end of reperfusion arteriolar diameter decreased by 1.8 ± 0.7% of the baseline, *p* < 0.01 vs. Hc, AngII:L-J groups. Microvascular leakage and ROS formation were prevented (NGL = 0.04 ± 0.01 and 0.05 ± 0.01, respectively, *p* < 0.01 vs. Hc, AngII:L-J groups) as well as the leukocyte adhesion (1 ± 1/100 μm v.l./30 sec, *p* < 0.01 vs. Hc, AngII:L-J groups). Moreover, the perfused capillary length was preserved (−5.8 ± 1.7% of the baseline, *p* < 0.01 vs. Hc, AngII:L-J groups)

### 3.5. Local Application of Candesartan Cilexetil Alone Did Not Significantly Reduce the Microvascular Damage Due to Hypoperfusion and Reperfusion

After local application, candesartan cilexetil did not significantly affect microvascular networks under baseline conditions. On the contrary, at the end of hypoperfusion pial arterioles were constricted by 20.5 ± 1.7% of the baseline (*p* < 0.01 vs. baseline and S group) (Figure 1), while microvascular permeability (Figure 2) and ROS formation (Figure 3) markedly increased with respect to the S group, reaching values similar to those observed in the Hc group (NGL = 0.21 ± 0.03 and 0.25 ± 0.02, *p* < 0.01 vs. baseline and S group).

After reperfusion arterioles remained constricted (diameter decreased by 14.8 ± 1.7% of the baseline, *p* < 0.01 vs. the baseline and S group) (Figure 1 and Figure 4(D1,D2)). The microvascular leakage further increased after reperfusion, reaching a value similar to that of the Hc Group (NGL = 0.42 ± 0.02, *p* < 0.01 vs. baseline and S group) (Figure 2 and Figure 4(D3)). ROS production remained similar to that measured after hypoperfusion (NGL = 0.30 ± 0.02, *p* < 0.01 vs. the baseline and S group) (Figure 3). Leukocyte adhesion increased (Figure 5), while capillary perfusion was reduced (Figure 6) with respect to the S group, with values that were not different compared to those measured in the Hc Group (8 ± 2/100 µm v.l./30 sec and PCL = −48 ± 4% of the baseline, *p* < 0.01 vs. the baseline and S group, respectively).

### 3.6. Local Application of Candesartan Cilexetil Prior to Local Angiotensin II Prevented Ang II Microvascular Damage Exacerbation

At the end of hypoperfusion, pial arterioles showed a diameter reduction; however, this reduction was lower than that measured for the AngII-L group (vessel diameter decreased by 16.8 ± 2.0% of the baseline, *p* < 0.01 vs. the baseline, S, Hc and AngII:L-J groups) (Figure 1). Both microvascular permeability (Figure 2) and ROS formation (Figure 3) increased with respect to the S group but was less than that of the AngII-L group (NGL = 0.10 ± 0.02 and 0.12 ± 0.02, *p* < 0.01 vs. baseline, S, Hc, AngII-L and C-L groups, respectively).

At the end of reperfusion, the diameter decreased by 12.5 ± 1.4% of the baseline (*p* < 0.01 vs. the baseline, S, Hc and AngII:L-J groups) (Figure 1). The microvascular leakage (Figure 2 and Figure 4(E1–E3)) and ROS production (Figure 3) were higher than that of the S group but lower than that of both Hc and AngII-L groups (NGL = 0.15 ± 0.03 and 0.08 ± 0.01, *p* < 0.01 vs. the baseline, S, Hc, AngII:L-J and C-L groups, respectively). With respect to the Hc and AngII-L groups, leukocyte adhesion was mostly prevented (5 ± 1/100 µm v.l./30 sec, *p* < 0.01 vs. S, Hc and AngII:L-J groups) (Figure 5), and capillary perfusion was partly preserved (PCL = -32.0 ± 2.5% of the baseline, *p* < 0.01 vs. the baseline, S, Hc, AngII:L-J and C-L groups) (Figure 6).

### 3.7. Jugular Infusion of Candesartan Cilexetil Protected the Pial Microcirculation from Damage Due to Hypoperfusion and Reperfusion

Candesartan cilexetil applied through jugular infusion was more effective than that applied via local application in attenuating microvascular changes. It prevented, indeed, the vasoconstriction observed after hypoperfusion (vessel diameter reduced by 2 ± 1% of the baseline, *p* < 0.01 vs. Hc, AngII:L-J and C-L groups) (Figure 1). Microvascular leakage (Figure 2) and ROS formation (Figure 3) were lower than that of both Hc and C-L groups (NGL = 0.12 ± 0.02 and 0.10 ± 0.01, *p* < 0.01 vs. S, Hc, AngII:L-J and C-L groups, respectively).

At the end of reperfusion, arterioles slightly dilated with respect to the baseline (by 4.6 ± 1.5% of the baseline, *p* < 0.01 vs. Hc, AngII:L-J and C-L groups) (Figure 1). Microvascular leakage (NGL = 0.14 ± 0.01, *p* < 0.01 vs. S, Hc, AngII:L-J and C-L groups) (Figure 2 and Figure 4(F1–F3)) and ROS formation (NGL = 0.06 ± 0.01, *p* < 0.01 vs. Hc, AngII:L-J and C-L groups) (Figure 3) were prevented by the leukocyte adhesion (2 ± 1/100 µm v.l./30 sec, *p* < 0.01 vs. Hc, AngII:L-J and C-L groups). The capillary perfusion was partially preserved (PCL = −33.0 ± 1.8% of the baseline, *p* < 0.01 vs. S, Hc, AngII:L-J and C-L groups).

### 3.8. Jugular Infusion of Candesartan Cilexetil, Prior to Intravenous Angiotensin II, Partially Protected the Pial Microvasculature from the Damage Induced by Hypoperfusion and Reperfusion

Candesartan cilexetil, after jugular administration, did not differ from its local application with respect to the vasoconstrictor effect. At the end of hypoperfusion, indeed, arterioles were constricted by 19.6 ± 1.5 % of the baseline (*p* < 0.01 vs. the baseline, S and AngII:L-J groups) (Figure 1). On the other hand, the microvascular leakage (Figure 2) and ROS formation (Figure 3) were reduced compared to the values in the Hc, AngII:L-J groups, although these effects were lower than in the C-AngII-L group (NGL = 0.28 ± 0.03 and 0.16 ± 0.01, *p* < 0.01 vs. baseline, S, AngII:L-J and C-AngII-L groups).

After reperfusion all arteriolar diameters moderately decreased, with an effect similar to that exerted by the local application of the drug (vessel diameters were reduced by 15.2 ± 1.8% of the baseline, *p* < 0.01 vs. the baseline, S, Hc, AngII:L-J groups) (Figure 1). Microvascular permeability (Figure 2 and Figure 4(G1–G3)) and ROS formation (Figure 3) were reduced with respect to the Hc, AngII:L-J groups, although to a lesser extent than they were in the C-AngII-L group (NGL = 0.36 ± 0.02 and NGL = 0.18 ± 0.02, *p* < 0.01 vs. the baseline, S, Hc, AngII:L-J and C-AngII-L groups, respectively). The number of adhered leukocytes (Figure 5) was reduced, and the capillary perfusion (Figure 6) was partially preserved with respect to the Hc and AngII-L groups, with an effect that was similar to that observed in the C-AngII-L group (6 ± 1/100 µm v.l./30 s, *p* < 0.01 vs. the baseline, S, Hc, AngII:L-J groups and PCL = −30.0 ± 3.5% of baseline, *p* < 0.01 vs. baseline, S, Hc, AngII:L-J groups, respectively).

### 3.9. Local Application of Mas Receptor Antagonist (A779) Prior to Local Ang 1-7 Abolished the Effects Induced by Ang 1-7

After local application of A779 and Ang 1-7, significant microvascular changes were not observedunder baseline conditions. At the end of hypoperfusion the arteriolar diameter decreased by 20.8 ± 2.4% of the baseline (*p* < 0.01 vs. the baseline and S group) (Figure 1), the microvascular leakage (Figure 2) and ROS formation (Figure 3) were significantly increased compared to the baseline (NGL = 0.24 ± 0.03 and 0.25 ± 0.02, respectively, *p* < 0.01 vs. the baseline and S group). After 60 min of reperfusione all arterioles were constricted compared with S group (−18.2 ± 1.8% of baseline, *p* < 0.01 vs. baseline and S group) (Figure 1). A greater increase in the microvascular permeability (NGL = 0.43 ± 0.02, *p* < 0.01 vs. baseline and S group) was seen compared to that observed at the end of hypoperfusion (Figure 2 and Figure 4(H1–H3)). The ROS formation was pronounced as well as the leukocyte adhesion to the venular wall (NGL = 0.30 ± 0.04 and 8 ± 2/100 µm v.l./30 s, *p* < 0.01 vs. S group) (Figure 3 and Figure 5). The capillary perfusion was impaired (PCL = −47 ± 3% of the baseline, *p* < 0.01 vs. the baseline and S group) (Figure 6).

### 3.10. Local Application of AT_2_R Antagonist (PD123319) Prior to Local Ang 1-7 Did Not Change the Effects Induced by Ang 1-7

The local application of PD1233319 and successively, that of Ang 1-7, did not significantly affect the pial microcirculation compared with baseline conditions. At the end of BCCAO arteriolar diameter decreased by 3.2 ± 1.8% of the baseline, which was not significant change compared to the S group (*p* < 0.01 vs. Hc and AngII:L-J groups) (Figure 1). Microvascular leakage (Figure 2) and ROS formation (Figure 3) were prevented (NGL = 0.04 ± 0.02 and 0.06 ± 0.03, respectively; *p* < 0.01 vs. Hc, AngII:L-J groups). At the end of reperfusion pial arterioles did not show significant changes compared with baseline conditions (diameter increase by 2.5 ± 0.8% of baseline; *p* < 0.01 vs. Hc, AngII:L-J groups) (Figure 1). Fluorescent dextran leakage (Figure 2 and Figure 4(I1–I3)) and ROS formation (Figure 3) were prevented (NGL = 0.06 ± 0.02 and 0.05 ± 0.01, respectively; *p* < 0.01 vs. Hc and AngII:L-J groups) as well as the leukocyte adhesion (2 ± 1/100 µm v.l./30 s, *p* < 0.01 vs. Hc, AngII:L-J groups). The perfused capillary length was preserved (PCL = −6.7 ± 1.5% of baseline, *p* < 0.01 vs. Hc, AngII:L-J groups) (Figure 6).

### 3.11. Jugular Infusion of A779 or PD123319 Prior to Intravenous Ang 1-7 Caused the Same Effects Observed with the Local Applications

A779 or PD123319 after jugular infusion did not cause different microvascular responses compared to those observed after their local application. In both experimental groups jugular infusion with A779 or PD123319 and intravenous infusion with Ang 1-7 did not induce any significantly alteration compared with baseline conditions. Jugular infusion with A770 prior to Ang 1-7 completely abolished the protective effect induced by jugular infusion with Ang 1-7. At the end of hypoperfusion, pial arterioles were constricted by 18.9 ± 1.5% of the baseline, (*p* < 0.01 vs. baseline and S group) (Figure 1). The increases in microvascular permeability and ROS formation were pronounced compared with the S group (NGL = 0.23 ± 0.01 and 0.27 ± 0.02, respectively; *p* < 0.01 vs. the baseline and S group) (Figure 2 and Figure 3). At the end of reperfusion the vasoconstriction still persisted (−17.5 ± 1.2% of the baseline, *p* < 0.01 vs. the baseline and S group) (Figure 1). Fluorescence leakage was pronounced as well as ROS formation (NGL = 0.47 ± 0.03 and 0.28 ± 0.02, *p* < 0.01 vs. the baseline and S group) (Figure 2 and Figure 3). The leukocyte adhesion was marked (10 ± 2/100 µm v.l./30 s, *p* < 0.01 vs. the baseline and S group) (Figure 5). The perfused capillary length was reduced by 43 ± 4% of the baseline (*p* < 0.01 vs. baseline and S group) (Figure 6). Jugular infusion with PD123319 prior to intravenous infusion with Ang 1-7 caused no significant decrease in arteriolar diameter (−2.8 ± 1.5% of baseline, *p* < 0.01 vs. Hc, AngII:L-J groups) at the end of hypoperfusion (Figure 1). Fluorescent dextran leakage and ROS formation were prevented (NGL = 0.06 ± 0.02 and 0.05 ± 0.01, respectively, *p* < 0.01 vs. baseline, Hc and AngII:L-J groups) (Figure 2 and Figure 3). At the end of reperfusion pial arterioles did not show any modification compared to baseline conditions (diameter increase by 3.8 ± 1.2% of aseline, *p* < 0.01 vs. Hc, AngII:L-J groups) (Figure 1). Microvascular leakage, ROS formation, leukocyte adhesion were prevented and capillary perfusion was preserved (NGL = 0.07 ± 0.02, NGL = 0.04 ± 0.01, 3 ± 1/100 µm v.l./30 sec, PCL = −5.3 ± 2.1% of the baseline, respectively, *p* < 0.01 vs. Hc and AngII:L-J groups) (Figure 2, Figure 3, Figure 5 and Figure 6).

### 3.12. Local Application of Diminazene Aceturate Prior to Ang II Prevented Ang II-Induced Microvascular Damage Exacerbation

After local application, diminazene aceturate did not significantly affect microvascular networks under baseline conditions. At the end of hypoperfusion pial arterioles showed only a slight reduction in diameter (vessel diameter decreased by 5.0 ± 1.8% of the baseline, *p* < 0.01 vs. the baseline, Hc, AngII:L-J groups) (Figure 1). Both microvascular permeability (Figure 2) and ROS formation (Figure 3) increased with respect to the S group, but remained lower compared to Hc, AngII:L-J groups (NGL = 0.09 ± 0.01 and 0.08 ± 0.02, *p* < 0.01 vs. the baseline, S, Hc, AngII:L-J groups, respectively). At the end of reperfusion the diameter of arterioles did undergo slight changes compared to the baseline (vessels diameter decreased by 6.8 ± 1.5% of the baseline; *p* < 0.01 vs. the baseline, S, Hc, AngII:L-J groups) (Figure 1). Microvascular leakage (Figure 2 and Figure 4(J1–J3)) and ROS production (Figure 3) were higher than that of the S group but lower than in the Hc and AngII:L-J groups (NGL = 0.11 ± 0.02 and 0.09 ± 0.01, *p* < 0.01 vs. the baseline, S, Hc, AngII:L-J groups, respectively). Leukocyte adhesion was mostly prevented (4 ± 2/100 µm v.l./30 s, *p* < 0.01 vs. the baseline, Hc, AngII:L-J groups) (Figure 5). Capillary perfusion was partly preserved (PCL = −27.6 ± 2.7% of the baseline, *p* < 0.01 vs. the baseline, S, Hc, AngII: L-J groups) (Figure 6).

### 3.13. Jugular Administration of Diminazene Aceturate Prevented the Microvascular Damage Due to Hypoperfusion and Reperfusion

At the end of hypoperfusion pial arterioles did not present a significant decrease in diameter, such as the reduction observed in the Hc, AngII:L-J groups (Figure 1). Microvascular leakage was almost prevented with respect to the Hc, AngII:L-J groups (NGL = 0.07 ± 0.01, *p* < 0.01 vs. S, Hc, AngII:L-J groups) (Figure 2), while ROS formation was prevented and reduced to levels that were lower than those of the S group (NGL = 0.03 ± 0.01, *p* < 0.01 vs. the baseline, Hc and AngII:L-J groups) (Figure 3). After reperfusion arterioles were dilated by 14.1 ± 2.2% of the baseline (*p* < 0.01 vs. the baseline, S, Hc and AngII:L-J groups) (Figure 1). Microvascular leakage (Figure 2 and Figure 4(K1–K3)) and ROS formation (Figure 3) were lower than those of the Hc, AngII:L-J groups (NGL = 0.09 ± 0.02, *p* < 0.01 vs. baseline, S, Hc, AngII:L-J groups and NGL = 0.05 ± 0.01, *p* < 0.01 vs. baseline, Hc, AngII:L-J groups, respectively). Leukocyte adhesion was prevented, and adhered leukocytes were similar in number to those detected in the S group (1 ± 1/100 µm v.l./30 s, *p* < 0.01 vs. the baseline and Hc, AngII:L-J groups) (Figure 5), while capillary perfusion was partially preserved (PCL = −24.5 ± 2.3% of the baseline, *p* < 0.01 vs. the baseline, S, Hc, AngII:L-J gropus).

The local administration of diminazene aceturate prior to hypoperfusion and at the beginning of reperfusion had the same microvascular effects as those observed after the diminazene aceturate injection into the jugular vein. After 30 min of hypoperfusion arteriolar diameter decreased by 3.2 ± 1.5% of the baseline (*p* < 0.01 vs. Hc and AngII:L-J groups). Microvascular leakage was prevented: NGL = 0.06 ± 0.02 (*p* < 0.01 vs. the Hc and AngII:L-J groups) as well as ROS formation: NGL = 0.02 ± 0.01 (*p* < 0.01 vs. the Hc and AngII:L-J groups). After reperfusion pial arterioles dilated by 12.8 ± 1.5% of the baseline (*p* < 0.01 vs. the baseline, S, Hc and AngII:L-J groups). Microvascular leakage and ROS formation were inhibited: NGL = 0.07 ± 0.01 and 0.03 ± 0.01, respectively (*p* < 0.01 vs. the Hc and AngII:L-J groups). Leukocyte adhesion was hindered: 2 ± 1/100µm v.l./30 s, vs. the Hc and AngII:L-J groups) and the capillary perfusion was partly preserved (−22 ± 3 of baseline, *p* < 0.01 vs. the baseline, S, Hc and AngII:L-J groups).

### 3.14. No differences between Sexes Were Observed in Terms of Microvascular Damage Induced by Hypoperfusion–Reperfusion

Finally, Ang II, either locally applied or intravenously infused in female rats, caused the same effects obtained in male rats. At the end of reperfusion, we observed vasoconstriction (by 21.2 ± 2.3% and 19.7 ± 1.3% of the baseline, respectively, *p* < 0.01 vs. the S and Hc groups). The fluorescent dextran leakage was pronounced compared to that of the Hc group (NGL = 0.56 ± 0.02 and 0.51 ± 0.01, respectively, *p* < 0.01 vs. the baseline, S and Hc groups) as well as the ROS formation (NGL = 0.49 ± 0.03 and 0.50 ± 0.02, respectively, *p* < 0.01 vs. the baseline, S and Hc groups). The leukocyte adhesion was marked (13 ± 2 and 11 ± 3/100 µm v.l./30 sec, respectively, *p* < 0.01 vs. the baseline and S group). Moreover, the capillary perfusion was impaired compared with the Hc group (PCL = −58 ± 4% and −63.0 ± 2.5% of the baseline, respectively, *p* < 0.01 vs. the baseline, S and Hc groups).

On the other hand, Ang 1-7, locally or intravenously administered, preserved pial microvasculature to the same extent as that detected in male rats. After 60 min of reperfusion no arteriolar diameters showed any significant difference with respect to baseline conditions (diameter increased by 1.8 ± 0.6% and 2.0 ± 0.7% of the baseline, respectively, *p* < 0.01 vs. the Hc and AngII:L-J groups). Permeability leakage was prevented (NGL = 0.04 ± 0.01 and 0.02 ± 0.01, respectively, *p* < 0.01 vs. the Hc and AngII:L-J groups) as well as ROS formation (NGL = 0.04 ± 0.02 and 0.03 ± 0.01, respectively, *p* < 0.01 vs. the Hc and AngII:L-J groups) and leukocyte adhesion was inhibited (2 ± 1 and 1 ± 1/100 µm v.l./30 s, respectively, *p* < 0.01 vs. the Hc and AngII:L-J groups). Finally, the capillaries were perfused quite completely (−3.2 ± 1.0% and −2.5 ± 1.2% of the baseline, respectively, *p* < 0.01 vs. the Hc and AngII:L-J groups).

## 4. Discussion

This study clearly indicates that Ang II significantly aggravated hypoperfusion and reperfusion microvascular damage via AT_1_R and oxidative stress, whereas Ang 1-7 prevented the damage in a model of rat pial microcirculation. Of note we chose a dose of Ang II as well as a dose of Ang 1-7 that did not affect microvascular functions under baseline conditions. However, the induction of cerebral blood flow hypoperfusion by BCCAO and subsequent reperfusion appeared to trigger Ang II-dependent effects, exacerbating microvascular changes when compared to those observed in hypoperfused and reperfused rats. Conversely, Ang 1-7 triggered protective effects, limiting redox stress and microvascular damage.

In the rat model used in this study, at the end of reperfusion Ang II caused massive microvascular leakage along the venular networks, causing interstitial edema and compromising the capillary viability, while ROS formation increased, indicating marked oxidative stress. Our data support previous studies that evaluated the effects caused by angiotensin II in different pathophysiological conditions, such as hypertension and hypercholesterolemia [34,35]. Ang 1-7 did not affect microvascular vessels under baseline conditions, but it was able to reduce extravascular leakage and counteract the increase in ROS. The role of angiotensin 1-7 on microcirculation during ischemia and reperfusion has been evaluated only in vitro in several experimental models. Our in vivo data support the protective effects on microvasculature observed in in vitro models [36,37]. These data indicate that the effects of Ang II and Ang 1-7 were opposite under hypoperfusion and reperfusion conditions, confirming the suggested competition between the RAS end final molecules, which are effective in exacerbating or preventing vascular damage under pathophysiological conditions, as also observed by Arroia and coworkers studying the different effects of Ang II and Ang 1-7 in ischaemic stroke [38].

Furthermore, the effects of Ang II were prevented by candesartan cilexetil, an antagonist of AT_1_R, which is known to induce arteriolar constriction and several proinflammatory effects [39]. It is interesting to highlight that candesartan cilexetil, locally applied, was not able to prevent or significantly reduce microvascular damage induced by hypoperfusion and reperfusion in the BCCAO rat model per se, which was accomplished in the absence of Ang II administration. However, candesartan cilexetil, when systemically administered, was effective in preventing damage induced by hypoperfusion and reperfusion in the present experimental model. Our data on the systemic effects triggered by i.v. candesartan cilexetil support previous results showing that candesartan was able to reduce infarct size in rats submitted to middle cerebral artery occlusion (MCAO) when administered prior to or after the occlusion of the cerebral artery [12]. It is worth noting that in our model the effects of candesartan cilexetil on pial arterioles were dependent on the route of administration, indicating that venous administration preserves pial microcirculation from hypoperfusion and reperfusion injury. Therefore, these data may suggest that angiotensin II plays a role in the cerebral damage induced by hypoperfusion and reperfusion injury. It was interesting to observe that the addition of Ang II was able to worsen microvascular damage and increase ROS production under hypoperfusion and reperfusion conditions in both male and female rats. Therefore, the protective effects of sex hormones do not appear to be involved in the microvascular damage caused by Ang II administration during hypoperfusion and reperfusion [40,41].

Data on the permeability increase induced by Ang II were reported in the in vitro model, such as the rat pulmonary microvascular endothelial cell monolayer [42]. Under stimulation with lipopolysaccharide (LPS), Ang II was able to further enhance the membrane permeability of this model [43]. Our data demonstrate that in the in vivo model Ang II could aggravate a permeability increase, suggesting that interstitial edema might be facilitated in conditions triggering the increase in Ang II levels. Or data support the observations that Ang II is able to stimulate adhesion molecules able to promote leukocyte rolling and adhesion to endothelial cells [37,44].

Ang 1-7 has been widely studied in the last two decades, demonstrating protection against cerebral ischemia induced in experimental models, such as the MCAO [45]. The effects were a reduction in infarct size and the promotion of angiogenesis, mainly mediated by the activation of Mas receptor, which is known to decrease inflammation cascades [46,47]. In our model, Ang 1-7 did not increase arteriolar diameter under baseline conditions (Figure 7).

During hypoperfusion and reperfusion, the main effects were the decrease in leakage and the adhesion of leukocytes, while the arteriolar diameter did not significantly increase compared to the baseline. At the end of reperfusion, however, the capillary perfusion was preserved, and there was reduction in ROS production. Our data support previous results, showing that Ang 1-7 did not increase cortical perfusion following reperfusion in a rat model of MCAO, even though there was an “indication to prevent the steel phenomenon in the contralateral hemisphere” [37]. On the other hand, Ang 1-7 was not able to reduce blood–brain barrier permeability immediately after reperfusion. In a rat model of Endothelin-1-induced MCAO, Ang 1-7 reduced the infarct size and attenuated the neurological deficits but did not alter the reduction in blood flow induced by Endothelin-1 administration [48]. Therefore, Ang 1-7 represents the second arm of the RAS, demonstrating effects opposite to those triggered by Ang II. Moreover, the present data indicate that the effects of Ang 1-7 during hypoperfusion and reperfusion were exclusively due to the activation of the MAS receptor. Its inhibition, indeed, completely abolished the protective effects of angiotensin 1-7, while the inhibition of the AT_2_R did not interfere with the protection triggered by Ang 1-7.

The effects of diminazene aceturate, both systemically or locally administered, indicate the key role of the ACE2 enzyme in balancing the overall RAS mechanisms, facilitating the catabolism of Ang II and the formation of Ang 1-7. By stimulating the activity of ACE2, diminazene aceturate was able to preserve pial microcirculation from damage induced by hypoperfusion and reperfusion in our model. The local effect triggered by diminazene aceturate was impressive, as it was able to protect pial microcirculation after local application of Ang II under the condition of hypoperfusion and reperfusion. Our data show that the local balance between ACE1 and ACE2 enzymes could play a key role in facilitating or inhibiting the effects of Ang II on microvascular networks under the conditions of hypoperfusion and reperfusion. Similar results were found by Malek and Nematbakhsh, who studied the effects of diminazene aceturate in the rat kidneys subjected to ischemia and reperfusion [49]. All the previous data [50,51,52,53] and our observations support the hypothesis that Ang II could play a role in inducing alterations in the microvascular networks of different organs. Conversely, Angiotensin 1-7 appears to trigger cascades of events preserving microvascular networks from damage.

Finally, it is worthwhile to consider the role of ACE1 and ACE2 in SARS-CoV2 infection. Taking into account the widespread presence of ACE1 and ACE2 in several organs, it has been hypothesized that the imbalance between the two arms of RAS may contribute to SARS-CoV2 infection. Oudit and coworkers, as well as Lanza and collaborators [54,55], indeed, suggested that SARS-CoV2 disease (COVID-19), through ACE2 involvement in the intracellular entry of the virus, may be effective in inducing a prevalence of Ang II compared to the amount of Ang 1-7. Consequently, the imbalance in ACE1 and ACE2 activity, with the accumulation of Ang II, may induce the activation of AT_1_R with consequent microvascular damage.

In conclusion, Ang II and Ang 1-7 are effective in inducing opposite effects in microvascular networks under hypoperfusion–reperfusion conditions. The imbalance between Ang II and Ang 1-7 could induce a prevalence of AT_1_R activation, which facilitates microvascular changes and redox stress with consequent organ implication. These pathways are under study to counteract the COVID-19 pandemic involving all the countries in the world, causing a high toll of human lives.

## Figures and Tables

**Figure 1 biomolecules-11-01861-f001:**
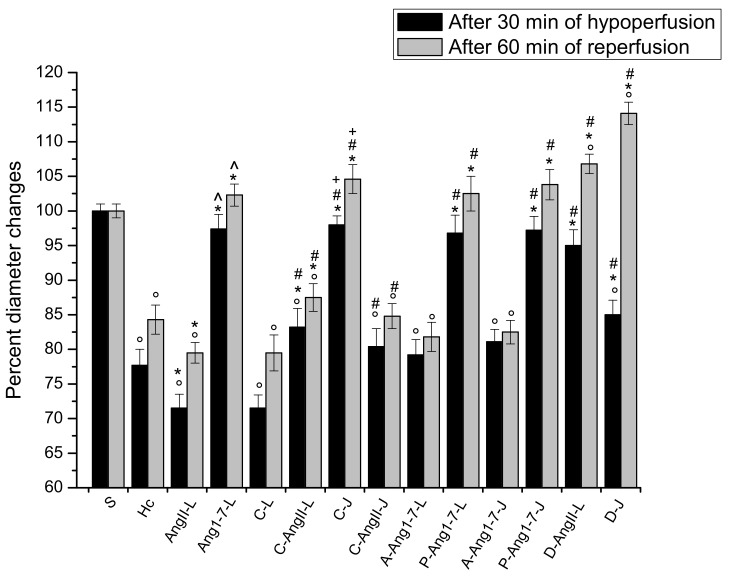
Diameter changes of order 3 pial arterioles. The data are expressed in percent of baseline diameter after 30 min of hypoperfusion and after 60 min of reperfusion in the different experimental groups: Sham-operated group (S group), hypoperfused group (Hc group), Ang II local application-treated group (AngII-L group), Ang 1-7 local application-treated group (Ang1-7-L group), candesartan cilexetil local application-treated group (C-L group), candesartan cilexetil plus Ang II local application-treated group (C-AngII-L group), candesartan cilexetil intravenous infusion-treated group (C-J group), candesartan cilexetil local application plus Ang II intravenous infusion-treated group (C-AngII-J group), A779 plus Ang 1-7 local application-treated group (A-Ang1-7-L group), P123319 plus Ang 1-7 local application-treated group (P-Ang1-7-L group), A779 plus Ang 1-7 intravenous infusion-treated group (A-Ang1-7-J group), P123319 plus Ang 1-7 intravenous infusion-treated group (P-Ang1-7-J group), diminazene aceturate plus Ang II local application-treated group (D-AngII-L group), and diminazene aceturate intravenous infusion-treated group (D-J group). All data are reported as mean ± SEM. ° *p* < 0.01 vs. S group, * *p* < 0.01 vs. Hc group, ^ *p* < 0.01 vs. AngII-L group, + *p* < 0.01 vs. C-L group and # *p* < 0.01 vs. AngII:L-J groups. Statistical significance was obtained using non-parametric test (Kruskal–Wallis test).

**Figure 2 biomolecules-11-01861-f002:**
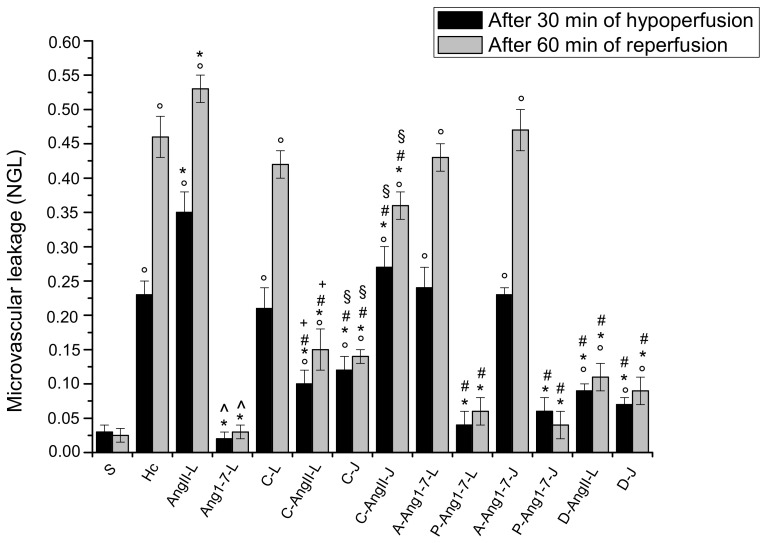
Microvascular permeability changes due to leakage of fluorescent dextran and expressed as normalized grey levels (NGL) in the different experimental groups: Sham-operated group (S group), hypoperfused group (Hc group), Ang II local application-treated group (AngII-L group), Ang 1-7 local application-treated group (Ang1-7-L group), candesartan cilexetil local application-treated group (C-L group), candesartan cilexetil plus Ang II local application-treated group (C-AngII-L group), candesartan cilexetil intravenous infusion-treated group (C-J group), candesartan cilexetil local application plus Ang II intravenous infusion-treated group (C-AngII-J group), A779 plus Ang 1-7 local application-treated group (A-Ang1-7-L group), P123319 plus Ang 1-7 local applied-treated group (P-Ang1-7-L group), A779 plus Ang 1-7 intravenous infusion-treated group (A-Ang1-7-J group), P123319 plus Ang 1-7 intravenous infusion-treated group (P-Ang1-7-J group), diminazene aceturate plus Ang II local application-treated group (D-AngII-L group), and diminazene aceturate intravenous infusion-treated group (D-J group). A marked increase in microvascular permeability compared to baseline conditions was observed in hypoperfused rats (Hc group) and those treated with local (AngII-L group) Ang II administration at the end of reperfusion. The local application of candesartan cilexetil prior to Ang II administration, (C-AngII-L and C-AngII-J group), prevents the increase in microvascular permeability as well as the local application of Ang 1-7 (Ang1-7-L group). The local or jugular vein infusion of Mas receptor antagonist prior to Ang 1-7 abolished the protective effect due to Ang 1-7, while the local or jugular vein infusion of AT_2_R antagonist prior to Ang 1-7 did not interfere with the protective effects due to Ang 1-7. Moreover, the jugular vein infusion of candesartan and cilexetil or diminazene aceturate prior to hypoperfusion and the beginning of reperfusion prevented the microvascular leakage (C-J and D-J group). The increase in microvascular permeability was also prevented by local application of diminazene aceturate prior to Ang II administration (D-AngII-L group). All data are reported as mean ± SEM. ° *p* < 0.01 vs. S group, * *p* < 0.01 vs. Hc group, ^ *p* < 0.01 vs. AngII-L group, + *p* < 0.01 vs. C-L group, # *p* < 0.01 vs. AngII:L-J groups and ^§^ *p* < 0.01 vs. C-AngII-L group. Statistical significance was obtained using parametric test (Bonferroni post hoc test).

**Figure 3 biomolecules-11-01861-f003:**
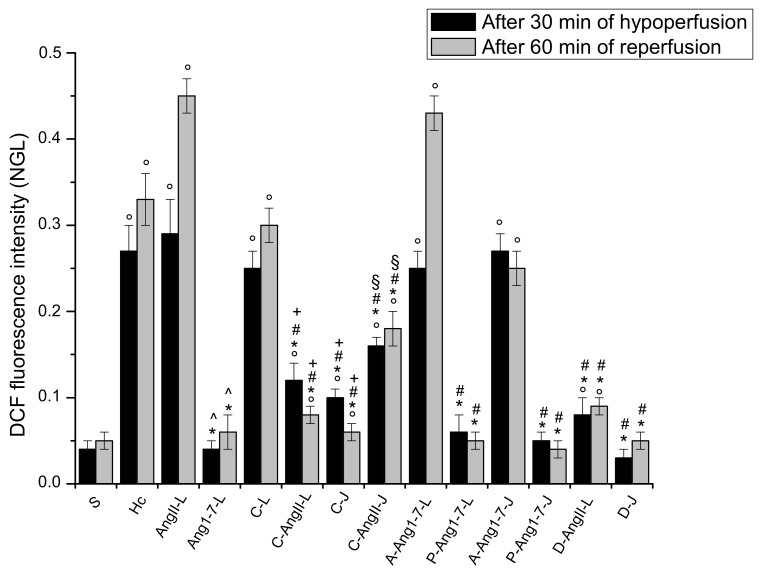
Changes in ROS production are related to the increase in DCF fluorescence intensity and are expressed as NGL at the end of hypoperfusion and at the end of reperfusion in the different experimental groups: Sham-operated group (S group), hypoperfused group (Hc group), Ang II local application-treated group (AngII-L group), Ang 1-7 local application-treated group (Ang1-7-L group), candesartan cilexetil local application-treated group (C-L group), candesartan cilexetil plus Ang II local application-treated group (C-AngII-L group), candesartan cilexetil intravenous infusion-treated group (C-J group), candesartan cilexetil local application plus Ang II intravenous infusion-treated group (C-AngII-J group), A779 plus Ang 1-7 local application-treated group (A-Ang1-7-L group), P123319 plus Ang 1-7 local applied-treated group (P-Ang1-7-L group), A779 plus Ang 1-7 intravenous infusion-treated group (A-Ang1-7-J group), P123319 plus Ang 1-7 intravenous infusion-treated group (P-Ang1-7-J group), diminazene aceturate plus Ang II local application-treated group (D-AngII-L group), and diminazene aceturate intravenous infusion-treated group (D-J group). The administration of AT_1_R antagonist or ACE II activator prior to Ang II administration reduced the ROS production compared with groups treated with Ang II alone as well as those administered with Ang 1-7. The administration of Mas receptor antagonist prior to Ang 1-7 did not prevent the ROS formation, while AT_2_R antagonist did not blunt Ang 1-7-induced prevention of ROS formation. Data are reported as Mean ± SEM. ° *p* < 0.01 vs. S group, * *p* < 0.01 vs. Hc group, ^ *p* < 0.01 vs. AngII-L group, + *p* < 0.01 vs. C-L group, and # *p* < 0.01 vs. AngII:L-J groups and ^§^ *p* < 0.01 vs. C-AngII-L group. Statistical significance was obtained using non-parametric test (Kruskal–Wallis test).

**Figure 4 biomolecules-11-01861-f004:**
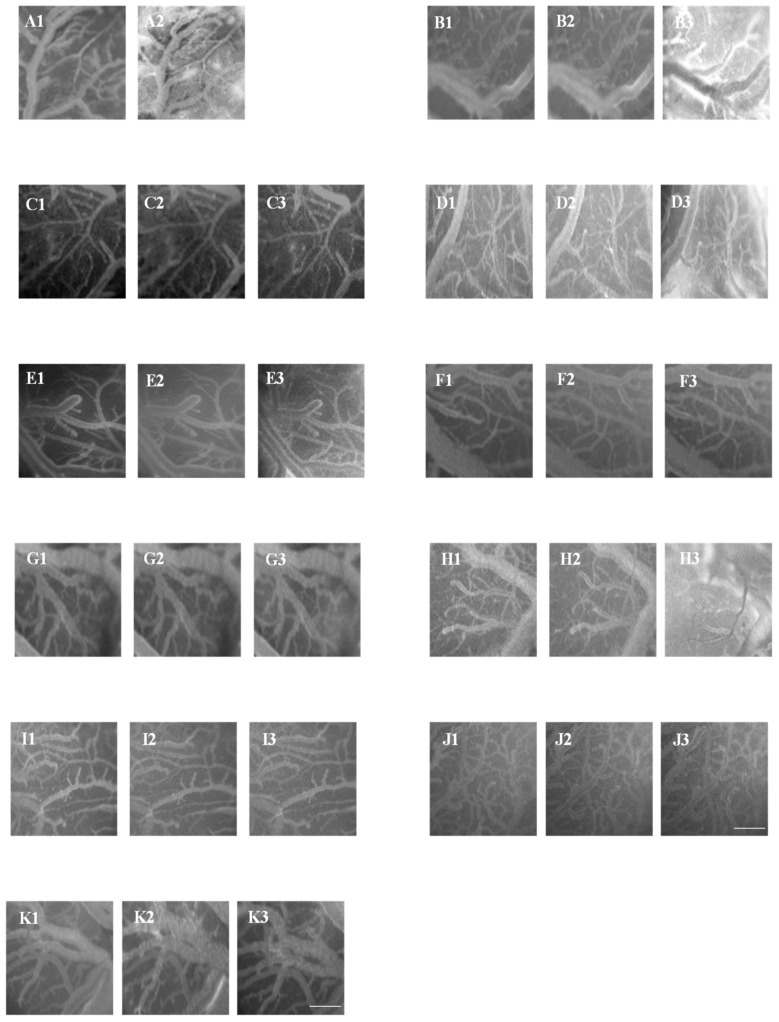
Images of pial microvascular networks obtained by computerized dedicated program. In (**A1**) the pial microvascular network is reported in baseline conditions (**A2**) after 60 min of reperfusion in a hypoperfused rat (Hc group), where it is possible to observe the increase in microvascular leakage by the marked change in the color of interstitium (from black to white). The pial network under baseline conditions (**B1**) 10 min after the local application of AngII (**B2**) and at the end of reperfusion (**B3**) in a rat belonging to the AngII-L group, where after local application of Ang II there was no change in microvascular permeability, while it significantly increased after reperfusion. (**C1**) reports the pial network under baseline conditions (**C2**) 10 min after the local application of Ang 1-7 and in (**C3**) after 60 min of reperfusion of a rat belonging to the Ang1-7-L group. No permeability increase was detected 10 min after Ang 1-7 administration or at the end of reperfusion. The pial network under baseline conditions (**D1**) 10 min after the local application of candesartan cilexetil (**D2**) and at the end of reperfusion (**D3**), where it is possible to evaluate a significant microvascular leakage as observed in the Hc group. (**E1**) reports the pial vasculature under baseline conditions (**E2**) 10 min after the local application of candesartan cilexetil plus Ang II locally applied and (**E3**) at the end of reperfusion of a rat of C-AngII-L group. No significant leakage of fluorescent dextran was detected compared with Hc group. (**F1**) reports the pial microcirculation under baseline conditions(**F2**) 10 min after the jugular vein infusion of candesartan cilexetil and (**F3**) at the end of reperfusion in a rat of C-J group. Image of the pial microvasculature under baseline conditions (**G1**) 10 min after the local application of candesartan cilexetil plus Ang II intravenously infused (**G2**) and at the end of reperfusion in a rat of C-AngII-J group (**G3**). (**H1**) shows the pial microvasculature under baseline conditions (**H2**) 10 min after the local application of Mas receptor antagonist plus Ang 1-7 locally applied and in (**H3**) at the end of reperfusion (A-Ang1-7-L group). It was detected significant fluorescence leakage as well as in Hc group. The pial network under baseline conditions (**I1**) 10 min after the local application of AT_2_R antagonist plus Ang 1-7 locally applied (**I2**) and at the end of reperfusion (**I3**) (P-Ang1-7-L group). No significant leakage of fluorescent dextran was observed compared with Hc group. Image of the pial microcirculation under baseline conditions (**J1**) after 10 min from the local application of diminazene aceturate plus Ang II locally applied (**J2**) and at the end of reperfusion (**J3**) of a rat of D-AngII-L group. (**K1**) reports the image of pial microcirculation under baseline conditions (**K2**) 10 min after the jugular vein infusion of diminazene aceturate and (**K3**) at the end of reperfusion in a rat of D-J group. In all four experimental groups no significant leakage of fluorescent dextran was detected 10 min after the different drug administrations and at the end of reperfusion compared with Hc group. Scale bar: 50 µm.

**Figure 5 biomolecules-11-01861-f005:**
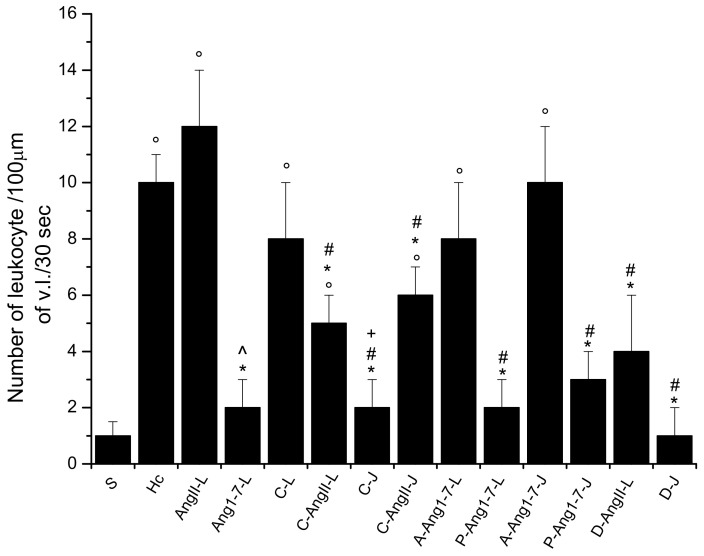
Evaluation of leukocytes sticking to the vessel walls over a 30 sec time period is expressed as the number of adhered cells/100 µm of venular length (v.l.)/30 sec at the end of reperfusion in the different experimental groups: Sham-operated group (S group), hypoperfused group (Hc group), Ang II local application-treated group (AngII-L group), Ang 1-7 local application-treated group (Ang1-7-L group), candesartan cilexetil local application-treated group (C-L group), candesartan cilexetil plus Ang II local application-treated group (C-AngII-L group), candesartan cilexetil intravenous infusion-treated group (C-J group), candesartan cilexetil local application plus Ang II intravenous infusion-treated group (C-AngII-J group), A779 plus Ang 1-7 local application-treated group (A-Ang1-7-L group), P123319 plus Ang 1-7 local applied-treated group (P-Ang1-7-L group), A779 plus Ang 1-7 intravenous infusion-treated group (A-Ang1-7-J group), P123319 plus Ang 1-7 intravenous infusion-treated group (P-Ang1-7-J group), diminazene aceturate plus Ang II local application-treated group (D-AngII-L group), and diminazene aceturate intravenous infusion-treated group (D-J group). Candesartan cilexetil prior to Ang II administration prevented leukocyte adhesion. Mas receptor antagonist prior to Ang 1-7 did not prevent leukocyte adhesion. Data are reported as mean ± SEM. ° *p* < 0.01 vs. S group, * *p* < 0.01 vs. Hc group, ^ *p* < 0.01 vs. AngII-L group, + *p* < 0.01 vs. C-L group and # *p* < 0.01 vs. AngII:L-J groups. Statistical significance was obtained using parametric test (Bonferroni post hoc test).

**Figure 6 biomolecules-11-01861-f006:**
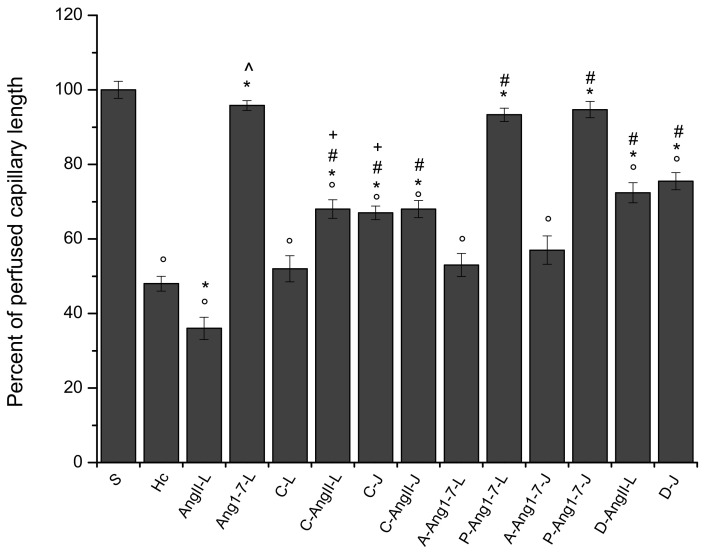
Changes in perfused capillaries are quantified as the length of the capillaries showing blood flow (PCL). Date are reported as percentage of the length of perfused capillaries at the end of reperfusion in the different experimental groups: Sham-operated group (S group), hypoperfused group (Hc group), Ang II local application-treated group (AngII-L group), Ang 1-7 local application-treated group (Ang1-7-L group), candesartan cilexetil local application-treated group (C-L group), candesartan cilexetil plus Ang II local application-treated group (C-AngII-L group), candesartan cilexetil intravenous infusion-treated group (C-J group), candesartan cilexetil local application plus Ang II intravenous infusion-treated group (C-AngII-J group), A779 plus Ang 1-7 local application-treated group (A-Ang1-7-L group), P123319 plus Ang 1-7 local applied-treated group (P-Ang1-7-L group), A779 plus Ang 1-7 intravenous infusion-treated group (A-Ang1-7-J group), P123319 plus Ang 1-7 intravenous infusion-treated group (P-Ang1-7-J group), diminazene aceturate plus Ang II local application-treated group (D-AngII-L group), and diminazene aceturate intravenous infusion-treated group (D-J group). Candesartan cilexetil local administration prior to Ang II preserved the capillary perfusion. Mas receptor antagonist prior to Ang 1-7 did not preserve the capillary perfusion, while AT_2_R antagonist did preserve capillary perfusion. Data are reported as mean ± SEM. ° *p* < 0.01 vs. S group, * *p* < 0.01 vs. Hc group, ^ *p* < 0.01 vs. AngII-L group, + *p* < 0.01 vs. C-L group and # *p* < 0.01 vs. AngII:L-J groups. Statistical significance was obtained using parametric test (Bonferroni post hoc test).

**Figure 7 biomolecules-11-01861-f007:**
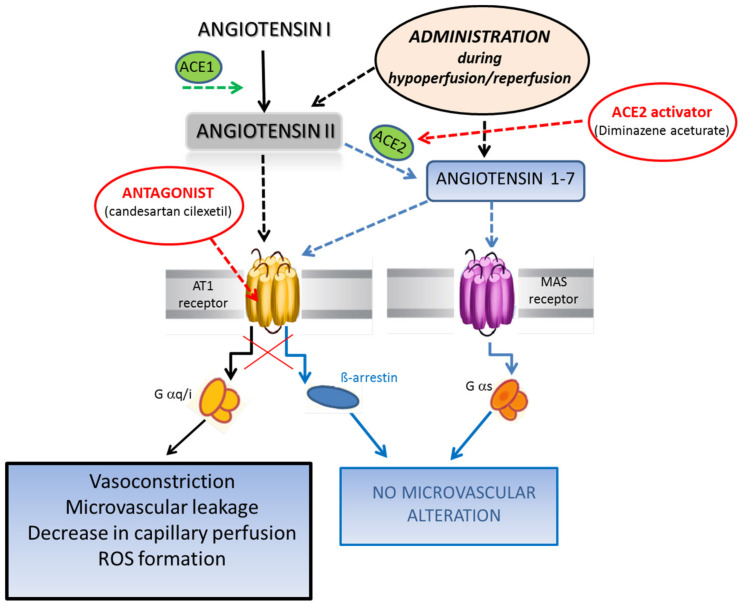
Summary of the crucial role of Ang II in microvasculature. The downregulation of ACE2 leads to an accumulation of Ang II to the detriment of Ang 1-7 formation. Ang II by binding mainly to AT_1_R triggers severe microvascular damage. The Ang II administration during hypoperfusion and reperfusion causes marked damage to the pial microcirculation, such as vasoconstriction, microvascular leakage, decrease in capillary perfusion and ROS formation, which are abolished through specific inhibition of AT_1_R by candesartan cilexetil. Conversely, Ang 1-7 administration or application of the ACE2 activator, diminazene aceturate, preserve the pial microcirculation from hypoperfusion and reperfusion injury.

**Table 1 biomolecules-11-01861-t001:** List of the different experimental groups with the relative treatments.

GROUPS	n	Sex	HYPOPERFUSION and REPERFUSION	TREATMENTS	
				Local Application	Intravenous Administration
**S**	5	M	no	no	no
**Hc**	11	M	yes	no	no
**AngII-L**	11	M	yes	Angiotensin II	no
**Ang 1-7-L**	11	M	yes	Angiotensin 1-7	no
**C-L**	11	M	yes	Candersartan cilexetil	no
**C-AngII-L**	11	M	yes	Candersartan cilexetil plus Angiotensin II	no
**A-Ang 1-7-L**	11	M	yes	A779 plus Angiotensin 1-7	no
**P-Ang 1-7-L**	11	M	yes	P123319 plus Angiotensin 1-7	no
**D-L**	11	M	yes	Diminazene aceturate	no
**D-AngII-L**	11	M	yes	Diminazene aceturate plus Angiotensin II	no
**AngII-J**	11	M	yes	no	Angiotensin II
**Ang 1-7-J**	11	M	yes	no	Angiotensin 1-7
**C-J**	11	M	yes	no	Candersartan cilexetil
**C-AngII-J**	11	M	yes	no	Candersartan cilexetil plus Angiotensin II
**A-Ang 1-7-J**	11	M	yes	no	A779 plus Angiotensin 1-7
**P-Ang 1-7**	11	M	yes	no	P123319 plus Angiotensin 1-7
**D-J**	11	M	yes	no	Diminazene aceturate
**D-AngII-J**	11	M	yes	no	Diminazene aceturate plus Angiotensin II
**AngII-LF**	11	F	yes	Angiotensin II	no
**Ang 1-7-LF**	11	F	yes	Angiotensin 1-7	no
**AngII-JF**	11	F	yes	no	Angiotensin II
**Ang 1-7-JF**	11	F	yes	no	Angiotensin 1-7

AngII:L-J indicates AngII-L group and AngII-J group.

**Table 2 biomolecules-11-01861-t002:** Classification in orders of pial arterioles according to Strahler’s method under baseline conditions.

ORDER	ARTERIOLES (n)	DIAMETER (μm)	LENGTH (μm)	RAT (n)
**5**	35	72.4 ± 4.5 *	987 ± 22	11
**4**	40	42.5 ± 3.3 *	763 ± 31	11
**3**	105	31.8 ± 2.0 *	1034 ±27	11
**2**	94	24.3 ± 1.2 *	880 ± 35	11
**1**	58	15.7 ± 0.7 *	310 ± 23	11

* *p* < 0.01 vs. different order.

## Data Availability

Not applicable.

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
