# Peer review of "The Effects of Angiotensin II or Angiotensin 1-7 on Rat Pial Microcirculation during Hypoperfusion and Reperfusion Injury: Role of Redox Stress"

_biomolecules, 2021, doi:10.3390/biom11121861_

Round 1
Reviewer 1 Report
I have some questions I wrote below
Local administration of candesartan alone (C-L) did not show a change in diameter, but local administration of angiotensin II and candesartan (CAII-L) increased the diameter (Figure 1). The author should discuss the reason in section of discussion.
The authors believe that angiotensin 1-7 acts on the MAS receptor, but the rationale is inadequate. The authors should add experimental data with the Mas receptor blocker A779. In addition, in considering the effect of angiotensin 1-7 on the angiotensin type 2 receptor, an experiment using the angiotensin type 2 receptor blocker PD123319 should be added.
In this study, diminazene aceturate was said to stimulate ACE2 activity, but data on whether ACE2 activity actually changed should be shown. This data should be shown.
Leaks are suppressed by angiotensin 1-7. However, angiotensin 1-7 may bind to angiotensin type 2 receptors and cause local kinin production. How do you consider that possibility?
Author Response
We thank the referee for his time and fruitful suggestions
We concur with the reviewer that the Figure 1 was in contrast with the real data reported in the paragraph concerning the effects of local administration of candesartan cilexetil prior to Angiotensin II. The results indicated that there was reduction in arteriolar diameter, as documented now in the revised Figure 1. We apologize for the mistake in reporting data in the previous Fig. 1.
We agree with the referee that the data on Angiotensin 1-7’s mechanisms of action through the MAS receptor and AT2R had to be assessed. Therefore, we carried out further experiments using the two specific inhibitors (A779: Mas receptor antagonist and P123319: AT2R antagonist) locally applied or injected into the jugular vein prior to Ang 1-7 administration.
In our experimental model, we observed that Angiotensin 1-7 was active exclusively on the Mas receptor: inhibiting this receptor with A779 we were able to blunt the protective effects of Ang 1-7; while the inhibition of the AT2R by P123319 did not change the preservation of microvascular networks triggered by Angiotensin 1-7. The results were reported in detail into the text (Results section: paragraphs 3.9, 3.10 and 3.11).
Diminazene aceturate has been proven to stimulate ACE2, with increase in formation of Angiotensin 1-7 compared to Angiotensin II. These effects have been reported in several experimental model (….). We administered diminazene aceturate both locally or intravenously injected prior to Angiotensin II during brain hypoperfusion and reperfusion injury. In our model, the effects of diminazene aceturate indicate that Angiotensin II, injected after diminazene, was not able to exacerbate the microvascular alterations, detected when we administered Angiotensin II both locally or systemically, by administration into the jugular vein. These effects have been reported in detail in the Results section (paragraphs: 3.12 and 3.13).
Moreover, we noted that the effects of locally applied diminazene aceturate were similar to those detected after injection of diminazene into the jugular vein, during brain hypoperfusion and reperfusion injury.
We have shown that Angiotensin 1-7 was active on the Mas receptor; several previous studies reported that the activation of this receptor facilitates the release of NO (Santos et al., Hypertension, 2003; Sampaio et al., Hypertension, 2007; Rukavina Mikusic et al. Exploration of Medicine, 2021). In our model, the inhibition of AT2R did not decrease the protective effects of Angiotensin 1-7. In particular, there was prevention of microvascular fluorescence leakage and reduction in ROS formation. Kinins have been reported to increase permeability of brain microcirculation and adhesion of leukocytes, causing cerebral oedema (Sikpa et al., Pharmaceuticals Basel 2020; Côté et al., Cancer Biol Ther. 2013). In our model we did not observe these effects after Angiotensin 1-7 administration or after inhibition of AT2R by antagonist P123319 administered prior to Angiotensin 1-7. We could hypothesize that kinins are not the prominent effector of Angiotensin 1-7 in our model.
Reviewer 2 Report
The present study investigated the effects of angiotensin II (AngII), angiotensin 1-7 (Ang-1-7) , candesartan and diminazene, either by topical or intravenous administration, on the various pathological aspects in brain of the hypoperfusion/reperfusion injury in the model of bilateral common carotid artery occlusion (BCCAO) in rats. The study demonstrated that AngII basically exacerbated the pathology by topical administration, while Ang-1-7 mitigated the pathology. Topical administration of candesartan had no effect on pathology, while it ameliorated the pathology observed with AngII. Topical application of diminazene prior to AngII prevented the pathogenic effect of AngII. Interestingly, venous administration of candesartan or diminazene alone significantly prevented the pathology. Accordingly,, authors propose that AngII and Ang-1-7 exacerbates and ameliorates, respectively, microvascular injury by acting through AT1R and Mas receptor.
Specific point
1. The involvement of Mas receptor in the action of Ang-1-7 remains to be established. The effect of Mas receptor antagonist, such as A779, on the mitigating effect of Ang-1-7 is recommended to be investigated. Otherwise, the proposal for the involvement of Mas receptor should be refrained.
2. In the Results, the experimental conditions are not accurately described. Precise description is recommended. For example, in lines 346-347, it is not clear and ambiguous that the results observed AngII were described, although it can be estimated from the context. The ambiguity is recommended to avoid in the scientific writing. Please scrutinize throughout the manuscript.
3. Not all data of 18 experimental groups are presented. "Data not shown" is recommended to avoid. The following list show some example. This can be done by showing the data in the supplemental information.
(1) Lines 339-340: The data supporing the statement "Ang II did not significantly affect microvascular networks under baseline conditions" are not presented.
(2) Section 3.4: it is recommended to show the results.
(3) Section 3.7: No data are show for D-L. Furthermore, the section title indicates that the section deals the effect of diminazene administration prior to AngII. However, as mentioned in the comment 2, there are some confusion and inaccuracy in the experimental condition in the writing of the results.
(4) The data obtained with female rats are recommended to show.
3. The observations with venous administration of candesartan alone can be taken as a significant results, which suggest that the hypoperfusion/reperfusion injury involves endogenous AngII-AT1R pathway. This observation is recommended to discusse and its importance can be emphasized in the Discussion. However, this comment is discretionary.
4. Lines 465-466: The statement "At the end of hypoperfusion pial arterioles did not present a significant decrease in diameter, such as the reduction observed in Hc and AngII-L groups" appears to be inconsistent with the results shown in Figure 1. Please clarify this.
5. Line 467: "abolished" is inaccurate. The microvascular leakage was prevented/inhibited but not abolished.
6. Line 489: "dramactically exacerbated" is kind of exaggerated expression and inaccurate.
Minor points
1. Is there any reason for setting glucose concentration of aCSF to 11 mM (200 mg/dL), which is un-physiological level.
2. Line 402: Please confirm the accuracy of figure "12.4+/-1.4%".
3. Line 479: "di" before "diminazene" is to be deleted.
Author Response
Reply
We thank the referee for his time and fruitful suggestions.
- We concur with the reviewer and we studied the effects of Mas receptor antagonist (A779) on the microvascular responses to Angiotensin 1-7. Therefore, we carried out further experiments using the two specific inhibitors (A779: Mas receptor antagonist) locally applied or injected into the jugular vein prior to Angiotensin 1-7 administration. In our experimental model, we observed that inhibition of Mas receptor caused significant decrease of the microvascular protection exerted by Angiotensin 1-7. These results were reported in detail into the text (Results section: paragraphs 3.9 and 3.11).
- We agree with the referee that the Results section was not adequately clear; therefore, we have revised the paper to improve the text, with a different sequence of the Results paragraphs and Figures.
- According to the referee’s suggestion, we deleted the words "data not shown". We added data as requested, but we tried to avoid a great length of Results section to facilitate the reading of the paper.
(1) We reported the data according to our observations. The diameter of arterioles did not change compared to baseline; there was no fluorescent leakage along the vessels; there was no adhesion of leukocytes to venular wall and all capillaries were perfused.
(2) We reported the data as requested.
(3) We revised the text clarifying the data.
(4) We showed the data obtained in the female rats. In particular, we reported the results at the end of reperfusion, resulting highly significant.
- We agree with the referee that the data concerning the candesartan cilexetil alone were really interesting, therefore we commented the data in the Discussion section (lines: 718-729).
- We concur with the referee that there was discrepancy between the data reported in the text and in figure 1; therefore, we corrected and revised Figure 1.
- We changed “abolished” with “prevented”
- We changed “dramatically exacerbated” with “significantly aggravated”
- The composition of artificial cerebrospinal fluid (aCSF) is fundamental in the brain to preserve the energy substrate and the typical aCSF formulations prescribe glucose concentration of 10–11 mM. This elevated concentration, compared to the physiological levels, is useful to protect the synaptic transmission against anoxic damage, as reported in several studies, and the powerful protection is due to the anaerobic metabolism of glucose and not to the higher osmolality, related to aCSF higher glucose.
- We confirm the accuracy of data reported in the text: 12.5 ± 1.4% with consequent correction of Figure 1.
- We deleted “di” before “diminazene”
The changes carried out into the text have been highlighted in red in the revised version
Round 2
Reviewer 1 Report
NO FURTHER comment.
Reviewer 2 Report
The manuscript has been satisfactorily revised and improved. This reviewer has no further comment.